# Identification of Potential Biomarkers and Biological Pathways for Poor Clinical Outcome in Mucinous Colorectal Adenocarcinoma

**DOI:** 10.3390/cancers13133280

**Published:** 2021-06-30

**Authors:** Chang Woo Kim, Jae Myung Cha, Min Seob Kwak

**Affiliations:** 1Department of Surgery, Ajou University College of Medicine, Suwon 16499, Korea; kcwgkim@gmail.com; 2Department of Internal Medicine, Kyung Hee University Hospital at Gangdong, Kyung Hee University College of Medicine, 892 Dongnam-ro, Gandong-gu, Seoul 05278, Korea; 4528359@gmail.com

**Keywords:** colorectal cancer, mucinous, survival, biomarker, gene

## Abstract

**Simple Summary:**

Patients with mucinous adenocarcinoma (MAC) have been considered to have a faster disease progression than patients with traditional adenocarcinoma (TAC) in colorectal cancer (CRC). However, to date, the roles of MAC in long-term survival remain controversial due to a small sample size and the nature of its relatively rare occurrence, although it potentially represents entities with different aggressiveness and prognoses. Here, using large-scale population data, we found that the patients with the MAC subtype had a significantly worse overall survival rate and a tendency of worse disease-specific survival rate in stage II compared with the patients with the TAC subtype. Furthermore, key gene signatures were identified using the established predictive models for the disease-specific survival of stage II mucinous CRC.

**Abstract:**

Colorectal cancer (CRC) comprises several histological subtypes, but the influences of the histological subtypes on prognosis remains unclear. We sought to evaluate the prognosis of mucinous adenocarcinoma (MAC), compared to that of traditional adenocarcinoma (TAC). This study used the data of patients diagnosed with CRC between 2004 and 2016, as obtained from the Surveillance, Epidemiology, and End Results database. We established a predictive model for disease-specific survival using conditional survival forest, model, non-linear Cox proportional hazards, and neural multi-task logistic regression model and identified the gene signatures for predicting poor prognosis based on the arrayexpress datasets. In total, 9096 (42.1%) patients with MAC and 12,490 (58.9%) patients with TAC were included. Those with the MAC subtype were more likely to have a poorer overall survival rate compared to those with the TAC subtype in stage II CRC (*p* = 0.002). The eight major genes including RPS18, RPL30, NME2, USP33, GAB2, RPS3A, RPS25, and CEP57 were found in the interacting network pathway. MAC was found to have a poorer prognosis compared to TAC, especially in Stage II CRC. In addition, our findings suggest that identifying potential biomarkers and biological pathways can be useful in CRC prognosis.

## 1. Introduction

Traditionally, the TNM staging system has played a fundamental role in the management of patients with colorectal cancer (CRC) as the most powerful and reliable predictor of prognosis [1]. However, clinicians often encounter various clinical outcomes after patients receive standard curative treatment, even among CRC patients with the same stage, which is unexpected and inexplicable. Increasing attention is being focused on the identification of new factors, which may enable a more accurate patient prognostic stratification within each stage in CRC.

Apart from the TNM staging system, histological classification, which is based on histological characteristics and genetic features traits, may influence the clinical features and outcome in CRC; thus, clarifying the effect of the varied histological subtypes will help clinicians choose the appropriate treatment strategy [2,3]. Mucinous adenocarcinoma (MAC) is defined by the World Health Organization (WHO) as ‘an adenocarcinoma containing more than 50% extracellular mucin within the tumor’ [4], which was found to occur in approximately 10% of all CRCs [5]. Compared with other subtypes, the MAC subtype is characterized by often occurring in a younger age [6], in a more advanced stage [7,8], and at the proximal colon [9,10]. From these findings, patients with MAC may be expected to have a faster disease progression than patients with traditional adenocarcinoma (TAC), which is classified as tubular, villous, or tubulovillous adenocarcinoma.

However, the actual clinical impact on the long-term survival of MAC patients compared that of with TAC patients and the causal link with its tumor biology remains highly controversial, although it potentially represents entities with different aggressiveness and prognoses [11]. To address these issues, we applied the incidence-based mortality method to Surveillance, Epidemiology, and End Results (SEER) data, a national population-based cancer registry [12], to investigate the long-term survival differences between MAC and TAC in CRC, followed by identifying the candidate biomarker genes and their pathways related to prognosis by investigating genomic data.

## 2. Materials and Methods

### 2.1. Data Sources

We used data from the National Cancer Institute’s SEER Program database, which was created to collect cancer incidence, prevalence, and survival data from U.S. cancer registries. SEER collects and publishes cancer incidence and survival data from 17 population-based cancer registries (Alaska Native tumor registry, Connecticut, Georgia Center for Cancer Statistics, San Francisco-Oakland, San Jose-Monterey, Greater California, Hawaii, Idaho, Iowa, Kentucky, Los Angeles, Louisiana, Massachusetts, New Mexico, New York, Seattle-Puget Sound, and Utah), covering approximately 34.6% of the U.S. population [13]. The database is broadly representative of the U.S. population. The SEER registry contains information on nine million cancer cases with over 550,000 new cases added to the database each year. It serves as a powerful resource for researchers focused on understanding the natural history of CRC and improving patient treatment [13,14]. This retrospective cohort study was reviewed and approved by the Institutional Review Board of the Kyung Hee University Hospital at Gangdong, Seoul, Republic of Korea (KHNMC IRB 2020-01-015). The need for informed consent was waived on account of the fact that all of the data used in this study were de-identified.

### 2.2. Study Population

The SEER registry collects data, including age at diagnosis, sex, race, primary site, histological type, tumor grade, tumor size, tumor depth, and survival. Using the SEER 1975–2016 database (released on 15 April 2019), we analyzed data from all of the patients diagnosed with CRC in the years 2004–2016. We extracted clinical and/or demographic data, including age at diagnosis, sex, race, and tumor information, including location, size, grade, histological type, and American Joint Committee on Cancer 7th TNM stages by using SEER disease codes. Tumor location was determined by using the following codes: cecum (C18.0), appendix (C18.1), ascending colon (C18.2), hepatic flexure (C18.3), transverse colon (C18.4), splenic flexure (C18.5), descending colon (C18.6), sigmoid colon (C18.7), overlapping lesion of colon (C18.8), colon (C18.9), rectosigmoid (C19.9), and rectum (C20.9). The morphology of cancer was categorized according to the third edition of the International Classification of Diseases for Oncology (ICD-O-3) histology and behavior codes: tubular adenocarcinoma, (8211/3); adenocarcinoma in villous adenoma (8261/3), villous adenocarcinoma (8262/3), adenocarcinoma in tubulovillous adenoma (8263/3), and MAC (8480/3). For tumor differentiation grading, we used a four-tier classification (well-differentiated, moderately differentiated, poorly differentiated, undifferentiated), which was proposed by the WHO grading system [15]. Patients were divided into two groups: the TAC group (patients with tubular adenocarcinoma, villous adenocarcinoma, and tubulovillous adenocarcinoma) and the MAC group (patients with MAC).

### 2.3. Establishment of Predictive Model

To develop a clinical prediction model for disease-specific survival, we used three predictive models for poor outcomes in CRC patients with MAC based on SEER data. For the tree approach, a conditional survival forest model (CSF) was conducted with 200 decision trees and a maximum depth of 5 to the square root of the total number of features [16]. The non-linear Cox proportional hazards (nCPH) model is frequently used to model survival data or time-to-event data, particularly in the presence of censored survival times [17]. Bent identity was used as an activation function. Furthermore, the learning rate, dropout rate, and number of epochs were set to 0.01, 0.2 and 5000, respectively. Finally, a neural multi-task logistic regression model (N-MTLR), which is based on a multi-task framework allowing the use of neural networks within the original MTLR design, was used [18]. The model was trained over 2000 epochs with a learning rate of 0.0001 using the Adam optimizer and rectified linear unit function. The patients were randomly assigned to a training set (80%) or a test set (20%).

### 2.4. Gene Signature Evaluation

To identify the different gene signatures for predicting poor outcome using the developed model, two previously published datasets (E-MTAB-863 and E-MTAB-864; ArrayExpress) were retrieved. A total of 37 patients from E-MTAB-863 and 18 patients from E-MTAB-864 were available for analysis after excluding patients with missing variables and other histological types except for MAC. The data were executed in robust multi-array analysis background correction and log2 transformed, then normalized through quantile normalization.

The linear models for the microarray data package in Bioconductor were utilized to mine statistically significant differentially expressed genes (DEGs) based on the difference in their expression values between MAC and TAC [19]. A *p*-value of <0.05 and |log2 fold change of | ≥ 1 were used as the cutoff criteria for this analysis. Further analysis, including construction of a human Protein-Protein Interaction (PPI) network, was carried out with the web-based tool NetworkAnalyst (http://www.networkanalyst.ca), which supports robust and reliable gene expression analysis [20,21], and pathway activations were selected and matched according to the Kyoto Encyclopedia of Genes and Genomes (KEGG) database.

### 2.5. Statistical Analysis

The SEER data were obtained using the SEER*Stat software (8.3.6 version; Surveillance Research Program, National Cancer Institute, Bethesda, MD, USA). To account for the potential effect of covariates on each outcome, we created a propensity score combining the covariates of age, sex, race, tumor grade, and tumor size and compared the prognosis of CRC patients with TAC and those with MAC. The cumulative probabilities of survival were assessed according to the Kaplan-Meier life-table analysis. A log-rank test was used to compare the survival rates between the TAC and MAC groups. Demographic differences between the two groups were tested using the Student’s *t*-test and Pearson chi-square test. All analyses were performed with Python (version 3.6.9) and R statistical software (version 3.6.0). A two-sided *p* ≥ 0.05 was considered statistically significant.

## 3. Results

### 3.1. Construction of the Prediction Model for Disease-Specific Survival

A total of 21,586 patients were included: 9096 (42.1%) patients with MAC and 12,490 (58.9%) patients with TAC in the SEER database. The MAC group had significantly higher proportions of older patients, female sex (except stage III), higher grades, and larger tumor sizes than the TAC group in all TNM stages (Appendix A). The propensity score matching eliminated significant differences between the two groups and the distribution of patient characteristics in each group was similar to that of all patients (Table 1).

After adjustment, we conducted subgroup analysis by tumor stage to further investigate the disease-specific and overall survival rates. The patients with the MAC subtype were more likely to have a poor overall survival rate compared to patients with the TAC subtype, notably patients with stage II (*p* = 0.002; Figure 1). It did not reach statistical significance for disease-specific survival between the two groups in stage II (*p* = 0.161). In addition, those in the MAC group showed poorer disease-specific survival rates than those in the TAC group with stage III and stage IV (*p* < 0.001 and *p* < 0.015; Figure 1). Similar results were also obtained by Cox proportional hazards analysis (Appendix A).

### 3.2. Development and Validation of a Risk Stratification Model for MAC with Stage II

Because we found a difference in the not-cause specific survival rate between the MAC and TAC groups in Stage II diseases, a disease-specific survival predictive model in Stage II CRC using CSF, n-CPH, and N-MTLR models were established (Figure 2). Tested on 20% of the SEER dataset using the predictive models, the root mean square errors (RMSEs) were 12.506, 26.714, and 11.529 in CSF, n-CPH, and N-MTLR, respectively. Similarly, the RMSEs were 6.775, 6.447, and 6.230 as a result of the test in the E-MTAB 863 and 864 test sets.

### 3.3. Identifying the Differentially Expressed Genes in ArrayExpress Dataset

Of the E-MTAB-863 and E-MTAB-864 datasets, a total of 34 patients, commonly classified into high-risk and low-risk groups from the three established machine learning models (CSF, n-CPH, and N-MTLR), were identified (Table 2). Of 117 DEGs, we identified the 12 DEGs (RPS18, USP33, CENPL, GAB2, RPS3A, RPS25, RPL30, HNMT, NME2, CEP57, ZC3H8, and TRIT1) most highly associated with disease-specific survival in stage II mucinous CRC (Figure 3, Appendix A). 

### 3.4. KEGG Pathway Analysis and Construction of the PPI Network

The interacting network analysis of responsive genes was further performed to identify the key genes. A total of 117 DEGs were subjected to KEGG pathway analysis (Appendix A). The genes were significantly enriched in the ‘pathways in cancer’, ‘viral carcinogenesis’, ‘PI3K-Akt signaling pathway’, ‘proteoglycans in cancer’, and ‘Ras signaling pathway’ (Appendix A). By performing further PPI network analysis, RPS18, RPL30, NME2, USP33, GAB2, RPS3A, RPS25, and CEP57 were indicated to be the common hub genes (Figure 4). Among them, RPS3A constituted the super hub node having the largest degree and highest betweenness (Appendix A).

## 4. Discussion

Using a large-scaled population data, we found that the patients with the MAC subtype had a significantly worse overall survival rate and a tendency of worse disease-specific survival rate in stage II compared with the patients with the TAC subtype. In addition, key gene signatures were identified using the established predictive models for disease-specific survival of stage II mucinous CRC.

Results of the prognosis of MAC compared with TAC are still conflicting. Several studies have reported no correlation of MAC with the oncologic outcomes of CRC patients. Purdie and Piris [22], Xie [23], and Warschkow et al. [24] did not find the presence of MAC to be an independent prognostic factor for overall and disease-specific survival in patients with CRC. Similarly, a study involving 1025 unselected patients from Italy showed that the overall survival of patients with stage II and III colon cancer with MAC was not significantly different from those with non-MAC [25]. One study even reported better overall survival rates in patients with MAC [26]. These discrepancies may reflect heterogeneity in the population of patients, lack of stratification by stage, exclusion of rectal primary tumors in some studies, and a small sample size.

In contrast, some authors reported that the MAC subtype was related with a poor prognosis [5,7,27,28,29]. Common explanations are that patients with MAC were younger and had more advanced tumor stages compared with those with TAC. Notably, in the stratified analysis according to stage from a large-scale population-based cohort, we found stage II in MAC is the clinically critical timepoint to have more aggressive behavior for survival than other histological subtypes. A meta-analysis showed that MAC patients had worse survival rates than TAC patients, even after the stage at diagnosis was similar in MAC and TAC patients in their meta-analysis [5]. The analysis showed a slightly worse prognosis for MAC (hazard ratio 1.05, 95% confidence interval 1.02–1.08) even when corrected for the stage at presentation. Another study with a stage-adjusted survival analysis from Germany also showed that the CRC patients with MAC with the same stage had a shorter disease-specific survival length and tended to have a worse disease-specific survival curve with a more advanced stage compared to those with TAC, but failed to achieve a level of statistical significance due to a small sample size [7].

However, even in the studies that showed a correlation of MAC with the prognosis, there are conflicts regarding MAC as an independent prognostic factor for CRC. A large retrospective study showed that MAC was not an independent prognostic factor of CRC in multivariate analysis, although the tumor stage and histological grade were higher in MAC than in TAC [7]. Similarly, another study reported that the MAC subtype was associated with adverse pathologic features, but it was not an independent prognostic factor [29]. These results suggest that other approaches, including biological, molecular, or genetic assessments, are needed to clarify the role of MAC in CRC.

Therefore, we tried to investigate novel insights into the biological differences associated with poor survival in stage II mucinous CRC. Of 12 DEGs for the disease-specific survival of stage II mucinous CRC, eight major genes with a biological relationship within the PPI network (RPS18, RPL30, NME2, USP33, GAB2, RPS3A, RPS25, and CEP57) were identified. 

Most of the genes have usually been investigated as prognostic biomarkers in renal cancer, and RPS18, RPL30, NME2, and RPS25 usually have been investigated as unfavorable predictors, while GAB2 has been reported as favorable predictor in renal cancer [30].

RPS18, the molecules within ribosomal proteins, has been demonstrated to be upregulated in CRC tissues. Its upregulation is a well-known common feature of active proliferation and the proliferation rate of tumor cells [31]. The increased number of ribosomal protein L30 (RPL30) may play a central role in the CRC process and in induced hepatocellular carcinoma [32,33]. The overexpression of NME2 was significantly associated with not only clinical parameters related to tumor progression, invasion, and metastasis but also resistance to 5-FU treatment [34,35]. The low expression of USP33 indicated a high recurrence risk and poor overall prognosis in advanced CRC patients [36]. Grb2-associated binders 2 (GAB2) contributed to tumor growth and angiogenesis through the upregulation of vascular endothelial growth factor expression, and it was also important in BRAF inhibition resistance [37,38]. RPS3A was determined to be a key gene associated with Microsatellite instability (MSI) as an important biological feature of CRC [39]. Although the association between RPS25, CEP57, and CRC has not yet been established, the relationship with other tumors has been reported. RPS25 was shown to be a potential biomarker of lung adenocarcinoma and adult T-cell leukemia [40,41], while CEP57 was associated with prostate cancer [42]. Taken together, with applications to CRC genomic data, the hub genes and networks for disease-specific survival of MAC in stage II revealed potential gene biomarkers. However, this study only represents the first step toward defining the roles of these genes in MAC, because there are only a few studies to date that investigate the roles of these genes. Therefore, additional validation, as well as more experimental research, is still required in order to verify the results of the present study.

Several limitations associated with our study should be noted. First, a lack of histological specificity from the ICD-O-3 codes in the SEER database may not precisely capture all of the patients. This, therefore, could lead to underestimating the true prevalence rate in TAC, rather than in MAC. As a result, due to the relatively higher proportion (42.1%) of MAC in the current study compared to previous reports, the disease-specific survival failed to confirm a statistical significance, despite the trend toward a worse outcome in stage II MAC. Second, its retrospective nature might contain bias despite propensity score-matching. Third is the lack of information on perioperative treatment or molecular tumor characteristics, such as the data on the RAS, BRAF, and MSI status, and serum CEA levels. Fourth, the mass data may reveal some confounders, for which no inter-observer agreement data were available for the assessment of MAC and TAC. In spite of these limitations, our study suggested three well-established predictive models and gene expression features to understand the MAC subtype based on the external validity and a high degree of power, along with minimizing potential confounding. With this, our study may contribute to a body of evidence that may identify reliable molecular biomarkers to more precisely stratify patients with MAC through the integrative analysis of large-scale clinical and gene expression data.

## 5. Conclusions

Our results indicate that the MAC subtype yielded worse overall and disease-specific survival compared with TAC subtype in stage II CRC and more advanced stages. The eight genes that were identified bear distinct cancer-specific attributes and, as a group, plays important roles in stage II mucinous CRC. Further studies validating the candidate gene biomarkers would be necessary to clarify the role of MAC in controlled clinical trials as well as in an experimental setting.

## Figures and Tables

**Figure 1 cancers-13-03280-f001:**
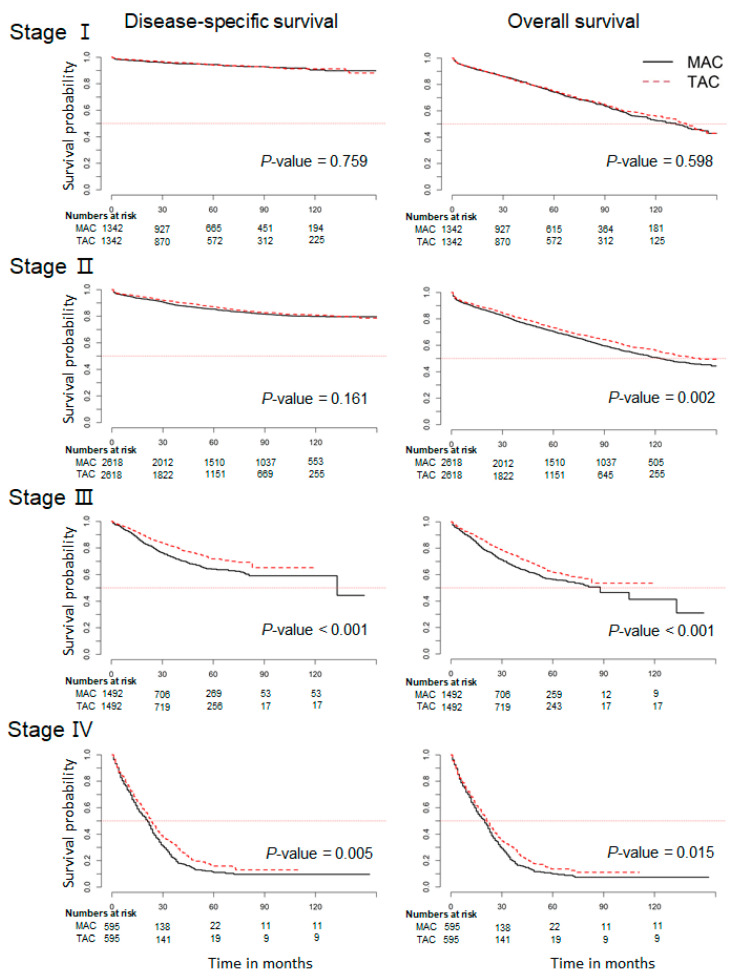
Kaplan–Meier curves of disease-specific and overall survival according to the stage between mucinous (MAC) and traditional colorectal cancer (TAC) in the SEER database.

**Figure 2 cancers-13-03280-f002:**
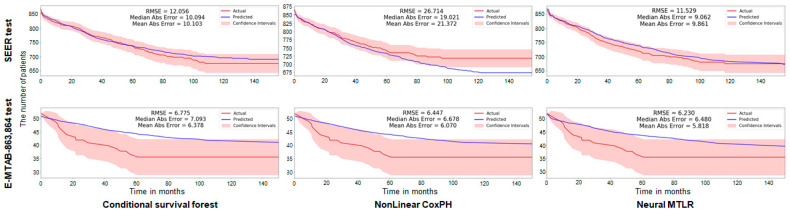
Internal and external validations of three developed models for disease-specific survival in mucinous colorectal cancer. The model shows reasonable discrimination in SEER database and arrayexpress datasets. RMSE, root mean square error.

**Figure 3 cancers-13-03280-f003:**
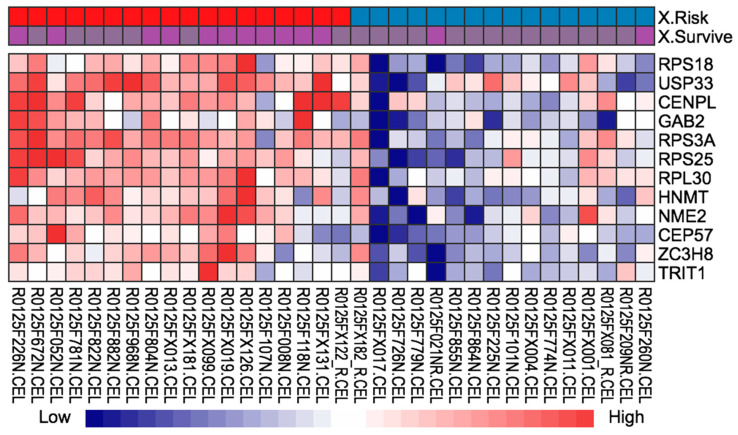
Differentially expressed genes identified from the arrayexpress datasets. Lower horizontal axis marks sample names, and right vertical axis represents gene names. Red represents up-regulated genes and blue represents down-regulated genes.

**Figure 4 cancers-13-03280-f004:**
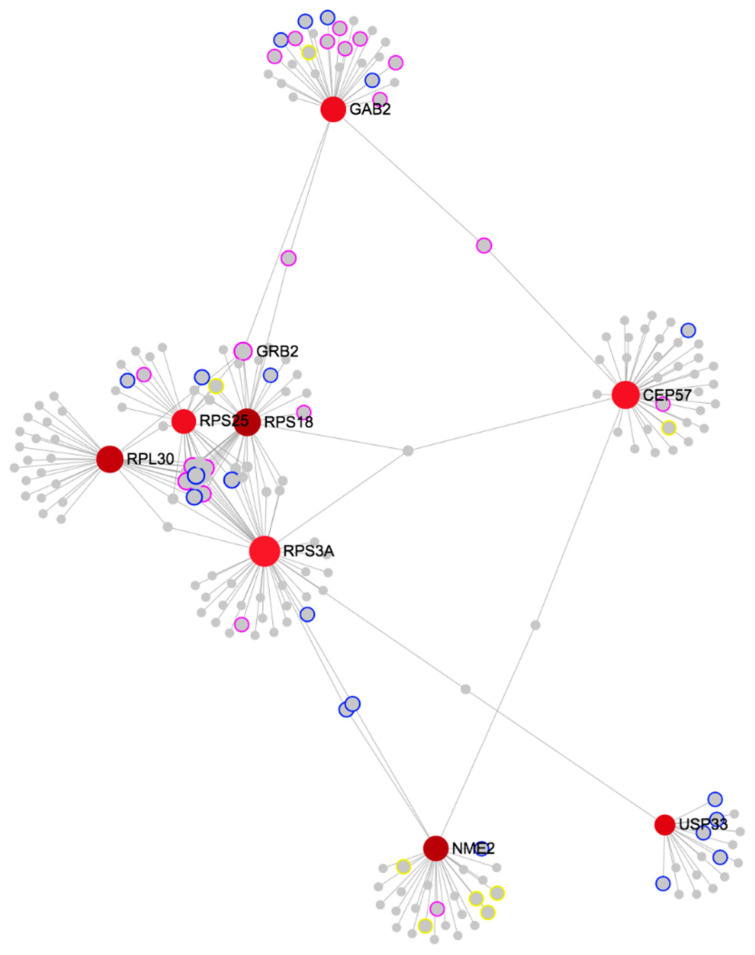
Protein-protein interaction networks of differentially expressed genes associated disease specific survival in mucinous colorectal cancer.

**Table 1 cancers-13-03280-t001:** Baseline characteristics of the patients with stage II CRC in SEER database.

KERRYPNX	Before Matching			After Matching		
	MAC *n* = 4390	TAC *n* = 2704	*p*-Value	MAC *n* = 2618	TAC *n* = 2618	*p*-Value
Age at diagnosis, y, mean (SD)	69.8 (14.5)	68.6 (13.9)	<0.001	68.8 (14.7)	68.9 (13.8)	0.877
Sex, *n* (%)			<0.001			0.934
Male	2010 (45.8)	1326 (49.0)		1267 (48.4)	1263 (48.2)	
Female	2380 (54.2)	1378 (51.0)		1351 (51.6)	1355 (51.8)	
Tumor grade, *n* (%)			<0.001			0.151
Well-differentiated	522 (11.9)	341 (12.6)		335 (12.8)	326 (12.5)	
Moderately-differentiated	3144 (71.6)	2105 (77.8)		2040 (77.9)	2036 (77.8)	
Poorly-differentiated	615 (14.0)	213 (7.9)		217 (8.3)	211 (8.1)	
Undifferentiated	109 (2.5)	45 (1.7)		26 (1.0)	45 (1.7)	
Primary site, *n* (%)			<0.001			
Cecum	1277 (29.1)	915 (33.8)				
Ascending	1219 (27.8)	605 (22.4)				
Hepatic flexure	302 (6.9)	141 (5.2)				
Transverse	531 (12.1)	252 (9.3)				
Splenic flexure	155 (3.5)	99 (3.7)				
Descending	217 (4.9)	129 (4.8)				
Sigmoid	592 (13.5)	512 (18.9)				
Rectal	97 (2.2)	51 (1.9)				
Race, *n* (%)			<0.001			0.917
Hispanic (All Races)	469 (10.7)	315 (11.6)		304 (11.6)	302 (11.5)	
Non-Hispanic American Indian/Alaska Native	12 (0.3)	10 (0.4)		8 (0.3)	7 (0.3)	
Non-Hispanic Asian or Pacific Islander	261 (5.9)	174 (6.4)		173 (6.6)	168 (6.4)	
Non-Hispanic Black	453 (10.3)	364 (13.5)		349 (13.3)	329 (12.6)	
Non-Hispanic White	3195 (72.8)	1841 (68.1)		1784 (68.1)	1812 (69.2)	
Tumor depth, *n* (%)			<0.001			
T3	3604 (82.1)	2376 (87.9)				
T4	786 (17.9)	328 (12.1)				
Tumor size, mm, mean (SD)	64.5 (42.4)	54.0 (47.6)	<0.001	54.3 (30.0)	53.3 (30.4)	0.211

MAC, mucinous adenocarcinoma; SD, standard deviation; SEER, Surveillance, Epidemiology, and End Results; TAC, traditional adenocarcinoma.

**Table 2 cancers-13-03280-t002:** Baseline characteristics of the patients with stage II CRC in array data.

Variables		High Risk	Low Risk	*p*-Value
		*n* = 18	*n* = 16	
Prognosis, *n* (%)				
	Good prognosis	5 (27.8)	13 (72.2)	0.006
	Poor prognosis	13 (72.2)	3 (16.7)	
Sex, *n* (%)				
	male	11 (61.1)	6 (33.3)	0.303
	female	7 (38.9)	10 (55.6)	
Cancer-related death, *n* (%)				
	Yes	11 (61.1)	2 (11.1)	0.011
	No	7 (38.9)	14 (77.8)	
Tumor grade, *n* (%)				
	Well-differentiated	1 (5.6)	2 (11.1)	0.466
	Moderately-differentiated	12 (66.7)	11 (61.1)	
	Poorly-differentiated	5 (27.8)	2 (11.1)	
	Undifferentiated	0 (0.0)	1 (5.6)	
Lymphovascular invasion, *n* (%)				
	Yes	5 (27.8)	2 (11.1)	0.507
	No	11 (61.1)	12 (66.7)	
	N/A	2 (11.1)	2 (11.1)	
T stage, *n* (%)				
	3	13 (72.2)	16 (88.9)	0.072
	4	5 (27.8)	0 (0.0)	
Tumor location, *n* (%)				
	Caecum	5 (27.8)	6 (33.3)	
	Ascending colon	1 (5.6)	8 (44.4)	0.017
	Hepatic flexure	2 (11.1)	0 (0.0)	
	Transverse colon	5 (27.8)	2 (11.1)	
	Splenic flexure	3 (16.7)	0 (0.0)	
	Descending colon	0 (0.0)	0 (0.0)	
	Sigmoid colon	2 (11.1)	0 (0.0)	
Family history of CRC				
	Yes	1 (5.6)	0 (0.0)	>0.999
	No	10 (55.6)	6 (33.3)	
	N/A	7 (38.9)	10 (55.6)	
Cancer recurrence				
	Yes	14 (77.8)	3 (16.7)	0.002
	No	4 (22.2)	13 (72.2)	
Age at diagnosis, y, mean (SD)		78.33 (8.02)	70.13 (5.82)	0.002
Length of follow up, y, mean (SD)		3.14 (2.16)	6.01 (2.74)	0.002
Tumor cell content, (%)		73.89 (14.61)	67.19 (16.22)	0.217
Number of regional LN assessed		14.5 (8.28)	17.5 (8.66)	0.312
Tumor size, cm, mean (SD)		7.02 (2.41)	5.28 (1.77)	0.022

CRC, colorectal cancer; LN, lymph node; SD, standard deviation.

## Data Availability

The data that support the findings of this study are openly available at https://seer.cancer.gov/ and http://www.ebi.ac.uk/arrayexpress (accessed on 30 November 2020).

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
