# Peer review of "Identification of Potential Biomarkers and Biological Pathways for Poor Clinical Outcome in Mucinous Colorectal Adenocarcinoma"

_cancers, 2021, doi:10.3390/cancers13133280_

Round 1
Reviewer 1 Report
The authors have seriously considered all the comments I have risen up and consequently improved the manuscript.
Reviewer 2 Report
Thanks for thoroughly assessing my concerns, I have no further issue with this manuscript.
This manuscript is a resubmission of an earlier submission. The following is a list of the peer review reports and author responses from that submission.
Round 1
Reviewer 1 Report
Identification of Potential Biomarkers and Biological Pathways for Poor Clinical Outcome in Mucinous Colorectal Adenocarcinoma
Chang Woo Kim, MD, PhD, Jae Myung Cha, MD, PhD, Min Seob Kwak, MD, PhD
The study by Kim et al. tries to evaluate whether mucinous colorectak adenocarcinoma has worse prognosis compared to conventional colorectal adenocarcinoma using large-scale population data (SEER). The team concludes that colorectal mucinous adenocarcioma has a significantly worse overall survival rate and a tendency of worse disease-specific survival rate in stage II compared with the patients with conventional colorectal adenocarcinoma.
The study is well written and the statistics seems sound.
I have one major concern with these findings (which admittedly has been only very briefly mentioned in the limitations) and this is the real “elephant in the room”. My concern is that no distinction between microsatellite unstable (MSI) and microsatellite stable (MSS) mucinous adenocarcinomas was made. This, I believe, is fundumental and is the main reason why previous efforts to identify a major aggressiveness to mucinous MSS have not produced such convincing results. If Kim et al were able to distinguish between these two cases and take into account differences between MSI mucinous and MSS mucinous it would lead to more robust results. This may in part explain why no differences are found in stage III cancers.
I am not sure if this is possible but, if it is not, it is definitaly a major limitation, indeed even more so than having no central histological revision which would ultimately be useful – especially with regards to grading etc.
Reviewer 2 Report
The submitted manuscript assigned cancers-1219015 by Kim et al. is dedicated to the identification of potential biomarkers and biological pathways for worse prognosis among CRC patients with mucinous CRC. This is well designed study, based on mining the data on a large, despite heterogenous, groups of CRC patients. The study will merit publication following corrections and responding the questions.
Regarding MAC and TAC, how are they related to molecular heterogeneity of CRC? Is for instance MAC otherwise homogenous?
Specific Comments: It would be good for readers if the authors would define TAC. Regarding localization for instance.
The most pronounced difference in prognosis concerns stage II. What do the authors suggest to the treatment of the patients regarding their findings? Not all TNM II patients undergo surgery and chemotherapy.
l.56: How about the MSI? This phenotype is of importance in proximal colon.
l.160-161: Once it is agreed on the significance threshold P=0.05, this is not significant. Please remove the text about tendency, since it is misleading.
Throughout the text the Figures and Tables would deserve better description.
l.164: To the Figure 1: Could the authors show the number of patients for each subchart?
l.201, Discussion: Epigenetic and genetic Mucin regulation was found in relation to CRC prognosis, please note following articles (Lu et al. PLOS One 2019, Vymetalkova et al. Carcinogenesis).
l.222, Table 1: Were all of them sporadic CRC patients?
Minor Comments: l. 49: genetic features traits, may influence the clinical features and…
l.51: …clinicians to choose…
l.59-62: Some sentences are rather long and therefore less comprehensible. The same occurs elsewhere in the text-please check.
l.90: Rephrase please. Clinical and/or demographic
Reviewer 3 Report
In this paper, using data obtained from the SEER database, Kim et al. describe that CRC patients with the MAC subtype have significantly poorer overall survival rate compared with TAC patients, and propose gene signatures as potential prognostic biomarkers. However, major and minor issues are present throughout this descriptive study.
Major
- The study lacks novelty and originality; previous studies have noted that patients with MAC have a poorer prognosis. The study is descriptive and requires substantial functional and mechanistic studies/data to fully interrogate the validity of the identified genes markers in appropriate CRC models. These analyses would be novel and exciting.
- The authors should refrain from stating throughout the manuscript that MAC showed ‘a tendency of worse disease-specific survival rate in stage II compared with the patients with the TAC subtype’. This wasn’t the case. The authors undertake these analyses using three models and in a very large subject number (n=20,000+); they yielded a non-significant p value (p>0.05) and shouldn’t over interpret any ‘trend’. Based on this non-significant result, they go on in the remainder of the paper to develop/validate a risk stratification tool for stage 2 MAC (and genes on the ArrayExpress). Thus the narrative of this study needs to be re-evaluated.
Minor
- Include a discussion on the identified gene(s) signatures in other neoplastic cancer models. Have they been identified previously in other GI models? Include any functional/mechanistic findings.
- Why was only 20% of the SEER database tested to develop/validate a risk stratification model for stage 2 MAC? Just clarify this. Clarify for non-experts the use of RMSEs here too.
- Include demographics of cohorts used for the gene signature evaluation and the ArrayExpress analyses.
- Include a hazard/risk ratio for all findings (any data where p<0.05).
- For the for ArrayExpress data, include a means which quantifies the strength of the genes’ association as being highly associated with disease-specific survival in stage 2.